# Deterministic Policy Gradient: Convergence Analysis

**Huaqing Xiong**[*1]     **Tengyu Xu**[*1]     **Lin Zhao**[2]     **Yingbin Liang**[1]     **Wei Zhang**[†3]

[1]Department of Electrical and Computer Engineering, The Ohio State University, Columbus, Ohio, USA
[2]Department of Electrical and Computer Engineering, National University of Singapore, Singapore, Republic of Singapore
[3]Department of Mechanical and Energy Engineering, Southern University of Science and Technology (SUSTech),
Shenzhen, Guangdong, China

## Abstract

The deterministic policy gradient (DPG) method proposed in Silver et al. [2014] has been demonstrated to exhibit superior performance particularly for applications with multi-dimensional and continuous action spaces. However, it remains unclear whether DPG converges, and if so, how fast it converges and whether it converges as efficiently as other PG methods. In this paper, we provide a theoretical analysis of DPG to answer those questions. We study the single timescale DPG (often the case in practice) in both on-policy and off-policy settings, and show that both algorithms attain an $\epsilon$-accurate stationary policy up to a system error with a sample complexity of $\mathcal{O}(\epsilon^{-2})$. Moreover, we establish the convergence rate for DPG under Gaussian noise exploration, which is widely adopted in practice to improve the performance of DPG. To our best knowledge, this is the first non-asymptotic convergence characterization for DPG methods.

## 1 INTRODUCTION

Reinforcement learning (RL) has achieved tremendous success so far in many applications such as playing video games [Mnih et al., 2013], bipedal walking [Castillo et al., 2019] and online advertising [Pednault et al., 2002], to name a few. The central aim of RL is to learn a policy that maximizes an accumulative reward for a task via the interaction with the environment. To this end, one popular method is to directly parameterize the policy and then optimize over the parameter space via (stochastic) gradient descent, which is referred to as the policy gradient (PG) algorithm [Williams, 1992]. More variants of policy gradient have been developed to further improve the performance, including natural policy gradient (NPG) [Kakade, 2002], trust region policy optimization (TRPO) [Schulman et al., 2015], proximal policy optimization (PPO) [Schulman et al., 2017], actor-critic (AC) Konda and Borkar [1999], Konda and Tsitsiklis [2000], Asynchronous Advantage Actor-Critic (A3C) [Mnih et al., 2016], Soft Actor-Critic (SAC) [Haarnoja et al., 2018], etc.

All the aforementioned PG algorithms adopt stochastic policies where the policy is modeled as a probability distribution over the action space. Rather, many RL applications have multi-dimensional **continuous** action spaces, for which **deterministic** policy gradient (DPG) algorithms have been proposed and demonstrated to significantly outperform stochastic PG algorithms in Silver et al. [2014]. Motivated by this, Lillicrap et al. [2016] combined DPG with DQN and proposed Deep Deterministic Policy Gradient (DDPG), which extends DQN in discrete action space to a continuous setting. Later, DDPG has also gained great success in distributional [Barth-Maron et al., 2018] and multi-agent [Lowe et al., 2017] scenarios. Although DPG and its variants exhibit superior performance in practice, the theoretical understanding of its convergence is rather limited. In fact, the only attempt was made in Kumar et al. [2020], which provided the convergence results for a modified zeroth-order DPG algorithm. However, what is used commonly in practice is the DPG algorithm originally proposed in Silver et al. [2014], for which the convergence guarantee remains open.

In fact, the convergence theory of DPG does not follow from that for stochastic PG algorithms due to a few unique features that DPG has. (a) The policy gradient in DPG takes a very different form from that in PG, and admits different compatibility for function approximation and consequently different actors to estimate. There is thus no guarantee that such a designed update rule must have guaranteed convergence as PG. (b) Through determinism in policy, practical implementation of DPG introduces **stochastic noisy** sampling to improve exploration. There is no previous theory on such a mixed deterministic policy update with noisy sampling for exploration. (c) DPG takes alternative simultaneous updates between critic and actor with constant learning

---

[*]Equal contribution
[†]Corresponding author

*Accepted for the 38th Conference on Uncertainty in Artificial Intelligence* (UAI 2022).

rates for both. Previous analysis for stochastic PG typically requires sufficiently fast update for critic so that its tracking error can be (asymptotically) decoupled from actor's convergence error. Such analysis is not applicable to DPG.

*Thus, the goal of this paper is to develop new tools to address the aforementioned challenges and provide the first finite-sample convergence guarantee for DPG algorithms in Silver et al. [2014].*

## 1.1 MAIN CONTRIBUTION

The main contribution of this work lies in establishing the first finite-sample analysis for both on-policy and off-policy DPG algorithms proposed in Silver et al. [2014].

For the on-policy setting, we study DPG-TD, which uses the compatible approximation rule given by Silver et al. [2014] to update actor, and adopts temporal difference (TD) learning with linear function approximation to update critic. We show that DPG-TD finds a stationary point (up to a system error) with a sample complexity of $\mathcal{O}(\epsilon^{-2})$. In addition, we also show that noised DPG-TD (NoiDPG-TD) which uses a mixture of noisy exploration and the deterministic policies can also achieve a sample complexity of $\mathcal{O}(\epsilon^{-2})$. For the off-policy setting, we study DPG-TDC, which uses TD with gradient correction (TDC) to update critic under off-policy data and show that DPG-TDC also achieves the convergence with a sample complexity of $\mathcal{O}(\epsilon^{-2})$.

Our results bring the following insights to the understanding of DPG. (a) Our sample complexity of DPG matches the best known actor-critic (AC) type PG in Xu et al. [2020b]. This implies that although the policy gradient of DPG is more challenging to estimate, the compatible function approximation for DPG in Silver et al. [2014] is as efficient as that for PG, which yields the same complexity for DPG. (b) DPG achieves the same sample complexity for more challenging continuous and possibly unbounded action space whereas the known theorems for PG typically require the bounded action space except for Gaussian policy. (c) The noisy exploration does not cause higher sample complexity. (d) The simultaneous updates for actor and critic without sufficient estimation accuracy for critic still yield convergence without causing more sample complexity.

Technically, our analysis develops the following novel techniques to handle the unique challenges arising due to deterministic policies. (a) We develop a new analysis to bound the estimation error of the Fisher information of deterministic policy arising via the compatibility property, and then further capture how such a metric affects the convergence via its minimum eigenvalue. (b) We develop a new tool to analyze the coupled actor and critic's stochastic approximation processes, due to their simultaneous updates both with constant stepsizes (which are commonly used in practice). Previous analysis of (stochastic) PG-type algorithms includ-

ing AC algorithms mainly decouples the critic's error from actor's error either by sufficient updates of critic before each actor's update [Wang et al., 2020, Kumar et al., 2019, Qiu et al., 2019, Xu et al., 2020b], or by updating critic much faster than actor via two timescale learning rates [Wu et al., 2020, Xu et al., 2020c]. Our analysis allows the coupling between the two and develops the idea to cancel the critic's coupling error by the actor's overall positive progress to the stationary policy.

## 1.2 RELATED WORK

**Convergence of DPG and its variants**: Since proposed and practically justified in Silver et al. [2014], DPG has inspired many variants and gained great success. However, there is almost no theoretical study of DPG with only one exception in Kumar et al. [2020], which provided the convergence guarantee for a zeroth-order DPG rather than the original form of DPG used widely in practice. Our work aims to provide the finite-sample convergence guarantee for DPG in its original practical form [Silver et al., 2014].

**Convergence of stochastic PG**: The vanilla PG is a fundamental policy-based RL algorithm. Its asymptotic convergence has been established in Williams [1992], Baxter and Bartlett [2001], Sutton et al. [2000], Kakade [2002], Pirotta et al. [2015], Tadić et al. [2017] via modeling PG as stochastic approximation (SA). PG has been shown to find the optimal policy under convex policy function approximation [Bhandari and Russo, 2019] or in some specific applications such as LQR [Fazel et al., 2018, Malik et al., 2019, Tu and Recht, 2019]. Convergence of (N)PG under a more general function approximation has been also provided in Shen et al. [2019], Papini et al. [2017, 2018, 2019], Xu et al. [2019, 2020a], Zhang et al. [2019], Agarwal et al. [2019], Karimi et al. [2019], Wang et al. [2020], Cen et al. [2020].

**Convergence of stochastic actor-critic**: The AC algorithm was proposed in Konda and Borkar [1999], and since then has aroused wide interest in understanding its convergence. Konda and Tsitsiklis [2000], Konda [2002], Peters and Schaal [2008], Bhatnagar et al. [2008, 2009], Bhatnagar [2010], Castro and Meir [2010], Maei [2018] established the asymptotic convergence for (natural) AC. The non-asymptotic convergence for (N)AC has been also explored recently. Under a double-loop setting, where critic can run sufficiently many iterations before updating actor, the convergence rate for (N)AC has been characterized in Yang et al. [2019], Wang et al. [2020], Kumar et al. [2019], Qiu et al. [2019], Xu et al. [2020b]. Under a two-timescale setting, where critic and actor update simultaneously but undergo different diminishing learning rates, the convergence rate has been provided in Wu et al. [2020], Xu et al. [2020c], Hong et al. [2020], Zhang et al. [2020], Shen et al. [2020]. Under the single timescale setting, where critic and actor

are simultaneously updated with constant learning rates, the convergence rate was given in Fu et al. [2021]. Differently from the above stochastic PG and stochastic AC, our work studies the deterministic PG algorithms. The update structure of our work is the same as that in Fu et al. [2021], but we adopt the more practical TD updates for critic rather than LSTD in Fu et al. [2021], which causes substantial difference in our analysis besides the deterministic policy.

## 2 PRELIMINARY

### 2.1 PROBLEM SETUP

We consider the standard RL settings where an agent interacts with a stochastic environment. Such a system is usually modeled as a discrete-time discounted Markov Decision Process (MDP) which is represented by a tuple $(\mathcal{S}, \mathcal{A}, P, r, \gamma)$. Here, $\mathcal{S}$ denotes the state space, $\mathcal{A}$ denotes the action space, $P : \mathcal{S} \times \mathcal{A} \times \mathcal{S} \mapsto [0, 1]$ denotes the transition kernel for the state transitions, e.g., $P(s'|s, a)$ represents the probability that the system takes the next state $s' \in \mathcal{S}$ given the current state $s$ and action $a$; $r : \mathcal{S} \times \mathcal{A} \mapsto [0, R_{\max}]$ is the reward function mapping the station-action pairs to a bounded subset of $\mathbb{R}$, and $\gamma \in (0, 1)$ is the discount factor.

Here, we consider a deterministic policy $\mu_\theta$ parameterized by $\theta \in \mathbb{R}^d$, namely, given the current state $s$, the policy follows a deterministic function mapping to generate an action $a = \mu_\theta(s)$. We also assume that the Markov chains generated by the policies are ergodic throughout this paper.

### 2.2 ON-POLICY DETERMINISTIC POLICY GRADIENT

We first consider the on-policy case where the interaction with the environment (i.e., the sampling) can follow the instantaneous target policy $\mu_\theta$.

The goal is to maximize the expected cumulative reward in the infinite-horizon case given by

$$J(\mu_\theta) = \int_{\mathcal{S}} \nu_{\mu_\theta}(s) r(s, \mu_\theta(s)) ds = \mathbb{E}_{s \sim \nu_{\mu_\theta}} [r(s, \mu_\theta(s))], \tag{1}$$

where $\nu_\mu(s') = \int_{\mathcal{S}} \sum_{t=0}^{\infty} \gamma^t p_0(s) p(s \to s', t, \mu) ds$ is the (improper) discounted state visitation distribution and $p(s \to s', t, \mu)$ denotes the density at state $s'$ after $t$ steps from state $s$ under policy $\mu$. In the remaining of this paper, we denote $J(\theta) := J(\mu_\theta)$ and $\nu_\theta := \nu_{\mu_\theta}$ for brevity.

One popular method to optimize the loss function defined in (1) is to use gradient-based algorithms such as stochastic gradient descent (SGD). To this end, the gradient of the loss function has been given by the so-called deterministic policy

gradient theorem [Silver et al., 2014] as follows:

$$\nabla J(\theta) = \int_{\mathcal{S}} \nu_\theta(s) \nabla_\theta \mu_\theta(s) \nabla_a Q^{\mu_\theta}(s, a)|_{a=\mu_\theta(s)} ds$$
$$= \mathbb{E}_{\nu_\theta} \left[ \nabla_\theta \mu_\theta(s) \nabla_a Q^{\mu_\theta}(s, a)|_{a=\mu_\theta(s)} \right]. \tag{2}$$

The policy gradient theorem for deterministic policies suggests a way to estimate the gradient via sampling, and then model-free policy gradient algorithms can be developed by following SGD updates for optimizing over policies. The difficulty of estimating the policy gradient $\nabla J(\theta)$ in (2) lies in approximating $\nabla_a Q^{\mu_\theta}(s, a)$. To address this difficulty, the compatible function approximation was established in Silver et al. [2014] which guarantees that $\nabla_a Q^{\mu_\theta}(s, a)$ can be replaced by $\nabla_a Q^w(s, a)$ in the policy gradient. We state such a property below, which is critical for designing the deterministic policy gradient algorithms.

**Proposition 1.** *(Compatible function approximation [Silver et al., 2014]) A function estimator $Q^w(s, a)$ compatible with a deterministic policy $\mu_\theta$, i.e., $\nabla J(\theta) = \mathbb{E}_{\nu_\theta} \left[ \nabla_\theta \mu_\theta(s) \nabla_a Q^w(s, a)|_{a=\mu_\theta(s)} \right]$, if it satisfies the following two conditions:*

1. $\nabla_a Q^w(s, a)|_{a=\mu_\theta(s)} = \nabla_\theta \mu_\theta(s)^T w$;
2. $w = w_{\xi_\theta}^*$ *minimizes the mean square error* $\mathbb{E}_{\nu_\theta} \left[ \xi(s; \theta, w)^T \xi(s; \theta, w) \right]$, *where* $\xi(s; \theta, w) = \nabla_a Q^w(s, a)|_{a=\mu_\theta(s)} - \nabla_a Q^{\mu_\theta}(s, a)|_{a=\mu_\theta(s)}$.

Following the compatibility property, the deterministic policy gradient can be rewritten as

$$\nabla J(\theta) = \mathbb{E}_{\nu_\theta} \left[ \nabla_\theta \mu_\theta(s) \nabla_\theta \mu_\theta(s)^T w_{\xi_\theta}^* \right], \tag{3}$$

where $w_{\xi_\theta}^*$ can be approximated easily by solving a regression problem.

### 2.3 OFF-POLICY DETERMINISTIC POLICY GRADIENT

In practice, it is often convenient to estimate the policy gradient via sampling under a behavior policy $\beta$, which is different from the target policy $\mu_\theta$. Silver et al. [2014] has also provided the deterministic policy gradient theorem for such an off-policy case, which is given by

$$\nabla J_\beta(\theta) = \int_{\mathcal{S}} \nu_\beta(s) \nabla_\theta \mu_\theta(s) \nabla_a Q^{\mu_\theta}(s, a)|_{a=\mu_\theta(s)} ds$$
$$= \mathbb{E}_{\nu_\beta} \left[ \nabla_\theta \mu_\theta(s) \nabla_a Q^{\mu_\theta}(s, a)|_{a=\mu_\theta(s)} \right], \tag{4}$$

where $\nu_\beta$ is the state visitation measure of the policy $\beta$. Correspondingly, the compatible form is given by

$$\nabla J_\beta(\theta) = \mathbb{E}_{\nu_\beta} \left[ \nabla_\theta \mu_\theta(s) \nabla_\theta \mu_\theta(s)^T w_{\beta, \xi_\theta}^* \right], \tag{5}$$

where $w_{\beta, \xi_\theta}^* = \arg \min_w \mathbb{E}_{\nu_\beta} \left[ \xi(s; \theta, w)^T \xi(s; \theta, w) \right]$ and $\xi(s; \theta, w)$ holds the same form as in Proposition 1.

# 3 ON-POLICY DPG ALGORITHM

In this section, we first describe the on-policy DPG algorithm proposed in Silver et al. [2014] and then provide the finite-sample convergence result for this algorithm.

## 3.1 ALGORITHM

In Silver et al. [2014], a compatible DPG algorithm using TD critic update was proposed, which we call as DPG-TD and describe in Algorithm 1. This algorithm introduces a critic parameter $w$ to estimate the gradient of Q-function based on the compatibility property. At each iteration, $w$ is updated by TD with a linear function approximator $Q^w(s,a) = \phi(s,a)^T w$ (line 9 of Algorithm 1). The algorithm uses $\theta$ as an actor parameter to update the policy (line 13 in Algorithm 1) based on the compatibility property.

---

**Algorithm 1** DPG-TD

---

1: **Input:** $\alpha_w, \alpha_\theta, w_0, \theta_0$, batch size $M$.
2: **for** $t = 0, 1, \ldots, T$ **do**
3:    **for** $j = 0, 1, \ldots, M - 1$ **do**
4:       Sample $s_{t,j} \sim d_{\theta_t}$. Generate $a_{t,j} = \mu_{\theta_t}(s_{,j})$.
5:       Sample $s_{t+1,j} \sim P(\cdot|s_{t,j}, a_{t,j})$ and $r_{t,j}$. Generate $a_{t+1,j} = \mu_{\theta_t}(s_{t+1,j})$.
6:       Denote $x_{t,j} = (s_{t,j}, a_{t,j})$.
7:       $\delta_{t,j} = r_{t,j} + \gamma \phi(x_{t+1,j})^T w_t - \phi(x_{t,j})^T w_t$.
8:    **end for**
9:    $w_{t+1} = w_t + \frac{\alpha_w}{M} \sum_{j=0}^{M-1} \delta_{t,j} \phi(x_{t,j})$.
10:   **for** $j = 0, 1, \ldots, M - 1$ **do**
11:      Sample $s'_{t,j} \sim \nu_{\theta_t}$.
12:   **end for**
13:   $\theta_{t+1} = \theta_t + \frac{\alpha_\theta}{M} \sum_{j=0}^{M-1} \nabla_\theta \mu_{\theta_t}(s'_{t,j}) \nabla_\theta \mu_{\theta_t}(s'_{t,j})^T w_t$.
14: **end for**

---

## 3.2 TECHNICAL ASSUMPTIONS

Before providing the result, we first introduce technical assumptions, all of which are standard or necessary mild regularity requirements.

**Assumption 1.** *For any $\theta_1, \theta_2, \theta \in \mathbb{R}^d$, there exist positive constants $L_\mu, L_\psi$ and $\lambda_\Psi$, such that (1) $\|\mu_{\theta_1}(s) - \mu_{\theta_2}(s)\| \leq L_\mu \|\theta_1 - \theta_2\|, \forall s \in \mathcal{S}$; (2) $\|\nabla_\theta \mu_{\theta_1}(s) - \nabla_\theta \mu_{\theta_2}(s)\| \leq L_\psi \|\theta_1 - \theta_2\|, \forall s \in \mathcal{S}$; (3) the matrix $\Psi_\theta := \mathbb{E}_{\nu_\theta} \left[ \nabla_\theta \mu_\theta(s) \nabla_\theta \mu_\theta(s)^T \right]$ (which we call as **Fisher information of deterministic policy**) is non-singular with the minimal eigenvalue uniformly lower-bounded as $\sigma_{\min}(\Psi_\theta) \geq \lambda_\Psi$.*

The first two statements can be easily satisfied for properly parameterized policy classes such as the linear approximator used in Silver et al. [2014] and the smooth neural network class. The last one ensures that $w^*_{\xi_\theta}$ defined in Proposition 1

is solvable and unique. A similar assumption has been also used in Liu et al. [2020].

**Assumption 2.** *For any $a_1, a_2 \in \mathcal{A}$, there exist positive constants $L_P, L_r$, such that (1) the transition kernel satisfies $|P(s'|s, a_1) - P(s'|s, a_2)| \leq L_P \|a_1 - a_2\|, \forall s, s' \in \mathcal{S}$; (2) the reward function satisfies $|r(s, a_1) - r(s, a_2)| \leq L_r \|a_1 - a_2\|, \forall s, s' \in \mathcal{S}$.*

In Assumption 2, the first statement is standard in the theoretical studies of RL [Bertsekas, 1975, Chow and Tsitsiklis, 1991, Dufour and Prieto-Rumeau, 2013, 2015], where the transition kernel is assumed to be Lipschitz continuous with respect to (w.r.t.) both state and action. Shah and Xie [2018] relaxes the Lipschitz continuity to be only w.r.t. state when considering a continuous state space. In this paper, we need the Lipschitz continuity to hold only w.r.t. action, because DPG algorithms are commonly used for continuous action space. The second statement can be easily satisfied for a properly defined reward function.

**Assumption 3.** *For any $a_1, a_2 \in \mathcal{A}$, there exists a positive constant $L_Q$, such that $\|\nabla_a Q^{\mu_\theta}(s, a_1) - \nabla_a Q^{\mu_\theta}(s, a_2)\| \leq L_Q \|a_1 - a_2\|, \forall \theta \in \mathbb{R}^d, s \in \mathcal{S}$.*

Assumption 3 indicates that the Q-function is smooth over action, which is a standard assumption in deterministic policy related studies [Kumar et al., 2020], and is also known as a principle to mitigate overfitting in the value estimation for actor-critic algorithms [Fujimoto et al., 2018].

**Assumption 4.** *The feature function $\phi : \mathcal{S} \times \mathcal{A} \to \mathbb{R}^d$ is uniformly bounded, i.e., $\|\phi(\cdot, \cdot)\| \leq C_\phi$ for some positive constant $C_\phi$. In addition, we define $A = \mathbb{E}_{d_\theta} \left[ \phi(x)(\gamma \phi(x') - \phi(x))^T \right]$ and $D = \mathbb{E}_{d_\theta} \left[ \phi(x) \phi(x)^T \right]$, and assume that $A$ and $D$ are non-singular. We further assume that the absolute value of the eigenvalues of $A$ are uniformly lower bounded, i.e., $|\sigma(A)| \geq \lambda_A$ for some positive constant $\lambda_A$.*

Assumption 4 is standard in the studies of TD learning with linear function approximation [Zou et al., 2019, Wu et al., 2020, Xu et al., 2020b,c]. This assumption guarantees the solvability of TD learning with linear function approximation. To be more specific, it ensures that $\mathbb{E}_{d_\theta}[\delta \phi] = 0$ has a unique root, namely $w^*_\theta$, which is also the global optimum of TD learning for a fixed policy $\mu_\theta$.

**Remark 1.** *We will abuse the notations a bit and assume that Assumptions 1 and 4 also hold for the off-policy case, with the expectations taken over the behavior stationary and (improper) visitation distributions $d_\beta$ and $\nu_\beta$.*

## 3.3 CONVERGENCE RESULT

In this subsection, we provide the finite-sample convergence analysis for DPG-TD in Algorithm 1.

Note that the deterministic policy gradient $\nabla J(\theta)$ in (2) has a different and more challenging form to analyze compared with stochastic PG, which requires the development of several new tools. First, we characterize the Lipschitz property for the deterministic policy gradient, which serves as a crucial step in the finite-sample analysis of DPG-TD. The previous study of DPG in Kumar et al. [2020] takes such a property as an assumption. Here, we formally establish such a Lipschitz property with the proof provided in Section 1 of the appendix, and characterize the dependence of the deterministic policy gradient on the basic parameters of the MDP.

**Lemma 1.** *Suppose Assumptions 1-3 hold. Then the deterministic policy gradient $\nabla J(\theta)$ defined in (2) is Lipschitz continuous with the parameter $L_J$, i.e., $\forall \theta_1, \theta_2 \in \mathbb{R}^d$,*

$$\|\nabla J(\theta_1) - \nabla J(\theta_2)\| \leq L_J \|\theta_1 - \theta_2\|, \qquad (6)$$

*where* $L_J = \left(\frac{1}{2}L_P L_\mu^2 L_\nu C_\nu + \frac{L_\psi}{1-\gamma}\right)\left(L_r + \frac{\gamma R_{\max} L_P}{1-\gamma}\right) + \frac{L_\mu}{1-\gamma}\left(L_Q L_\mu + \frac{\gamma}{2}L_P^2 R_{\max} L_\mu C_\nu + \frac{\gamma L_P L_r L_\mu}{1-\gamma}\right).$

In the following, we provide the convergence guarantee for DPG-TD. Our main technical novelty lies in the development of a new framework to analyze the **coupled** actor and critic's stochastic approximation processes, due to their simultaneous updates both with constant stepsizes. Our central idea is to cancel the critic's cumulative coupling error by the overall positive progress of actor's approach to the stationary policy. This is different from the previous analysis of (stochastic) PG-type algorithms which mainly decouples or asymptotically decouples the critic's error from actor's error. We provide a proof sketch in Section 3.4 with the full proof given in Section 2 of the appendix.

**Theorem 1.** *Suppose that Assumptions 1-4 hold. Let $\alpha_w \leq \frac{\lambda}{2C_A^2}; M \geq \frac{48\alpha_w C_A^2}{\lambda}; \alpha_\theta \leq \min\left\{\frac{1}{4L_J}, \frac{\lambda\alpha_w}{24L_h L_w}\right\}$. Then the output of DPG-TD in Algorithm 1 satisfies*

$$\min_{t\in[T]}\mathbb{E}\|\nabla J(\theta_t)\|^2 \leq \frac{c_1}{T} + \frac{c_2}{M} + c_3\kappa^2,$$

*where* $c_1 = \frac{8R_{\max}}{\alpha_\theta(1-\gamma)} + \frac{144L_h^2}{\lambda\alpha_w}\|w_0 - w_{\theta_0}^*\|^2, c_2 = \left[48\alpha_w^2(C_A^2 C_w^2 + C_b^2) + \frac{48L_w^2 L_\mu^4 C_{w_\xi}^2 \alpha_\theta^2}{\lambda\alpha_w}\right] \cdot \frac{144L_h^2}{\lambda\alpha_w} + 36L_\mu^4 C_{w_\xi}^2, c_3 = 18L_h^2 + \frac{24L_\mu^2 L_h^2 \alpha_\theta^2}{\lambda\alpha_w}$ *with* $C_A = 2C_\phi^2, C_b = R_{\max}C_\phi, C_w = \frac{R_{\max}C_\phi}{\lambda_A}, C_{w_\xi} = \frac{L_\mu C_Q}{\lambda_\Psi(1-\gamma)}, L_w = \frac{L_J}{\lambda_\Psi} + \frac{L_\mu C_Q}{\lambda_\Psi^2(1-\gamma)}\left(L_\mu^2 L_\nu + \frac{2L_\mu L_\psi}{1-\gamma}\right), L_h = L_\mu^2, C_Q = L_r + L_P \cdot \frac{\gamma R_{\max}}{1-\gamma}, L_\nu = \frac{1}{2}C_\nu L_P L_\mu$, *and $L_J$ defined in Lemma 1, and we define*

$$\kappa := \max_\theta \|w_\theta^* - w_{\xi_\theta}^*\|. \qquad (7)$$

Theorem 1 indicates that the convergence upper bound consists of three parts. The first term captures the convergence

rate and vanishes sublinearly with the number of iterations. The second term captures the variance caused by the stochastic sampling and can be controlled by the batch size. The last term captures the system error $\left\|w_\theta^* - w_{\xi_\theta}^*\right\|$ which is uniformly bounded by some constant $\kappa$. Such a system error includes two parts of the approximation errors. The first part is introduced by the difference between the optimal output Q-function of TD learning and the ground truth Q-function. The second part captures the approximation error due to the fact that none of the linear functions in this class satisfies the compatibility property in Proposition 1. In practice, the high capacity of the neural network class can significantly help to reduce such an error and achieves better convergence accuracy.

Theorem 1 also captures how the Fisher information of deterministic policy $\Psi_\theta := \mathbb{E}_{\nu_\theta}\left[\nabla_\theta\mu_\theta(s)\nabla_\theta\mu_\theta(s)^T\right]$ affects the convergence rate via its minimum eigenvalue bound $\lambda_\Psi$. Such a metric arises due to the compatible function approximation and captures how well actor estimates the deterministic policy gradient. Clearly, larger $\lambda_\Psi$ indicates a better system condition (smaller $C_{w_\xi}$ and $L_w$) and hence a faster convergence.

Based on the convergence rate in Theorem 1, we provide the sample complexity of the algorithm as follows.

**Corollary 1.** *Suppose that the same assumptions in Theorem 1 hold. Then the output of DPG-TD in Algorithm 1 satisfies $\min_{t\in[T]}\mathbb{E}\|\nabla J(\theta_t)\|^2 \leq \epsilon + c_3\kappa^2$, by using the total number of samples $2MT = \mathcal{O}\left(1/\epsilon^2\right)$.*

Corollary 1 shows that DPG-TD attains an $\epsilon$-accurate stationary point (up to the system error) with a sample complexity of $\mathcal{O}(\epsilon^{-2})$. To our best knowledge, this is the first finite-sample characterization for DPG.

Despite the policy gradient of DPG is more challenging to estimate, the sample complexity of DPG in Corollary 1 matches the best known stochastic PG (with AC scheme) in Xu et al. [2020b]. Furthermore, such a result does not require critic's update in DPG to accurately track the deterministic policy gradient at each step and hence is practically desired, whereas the sample complexity guarantee in Xu et al. [2020b] for stochastic PG requires sufficient accuracy of tracking the policy gradient at each iteration.

**DPG with noisy sampling for exploration.** In practice, the deterministic policy used in Algorithm 1 usually suffers from the inefficient exploration. To overcome such an issue, Silver et al. [2014] proposed to use a noisy sampling for DPG. To be specific, in lines 4-5 of Algorithm 1, a noisy policy, e.g., $\pi_{\theta_t}(s) = \mu_{\theta_t}(s) + \mathcal{N}(0, \sigma^2)$, is adopted to generate actions $a_{t,j}, a_{t+1,j}$. Correspondingly, the states for critic's updates in lines 4-5 of Algorithm 1 are generated by the stationary distribution $d_{\pi_{\theta_t}}$ associated with the noisy policy $\pi_{\theta_t}$. The rest of Algorithm 1 is unchanged. We refer

to such an algorithm as noisy DPG-TD (NoiDPG-TD).

Following the same techniques for the proof of Theorem 1 and replace the stationary distribution $d_{\theta_t}$ by $d_{\pi_{\theta_t}}$, we readily obtain the convergence result for NoiDPG-TD as follows.

**Corollary 2.** *Suppose that the same assumptions in Theorem 1 hold. Then the output of NoiDPG-TD satisfies $\min_{t \in [T]} \mathbb{E} \left\| \nabla J(\theta_t) \right\|^2 \leq \epsilon + \mathcal{O}\left(\kappa^2\right)$, by using the total number of samples $\mathcal{O}\left(\epsilon^{-2}\right)$.*

In Corollary 2, the system error $\kappa$ is determined by the noisy policy. Corollary 2 indicates that the noisy sampling for exploration does not cause higher sample complexity compared to DPG-TD.

### 3.4 PROOF SKETCH OF THEOREM 1

In the following, we outline the proof of Theorem 1 to highlight our new approach to analyzing the **coupled** actor and critic's stochastic approximation processes, due to their simultaneous updates both with constant stepsizes. The central idea is to cancel the critic's cumulative tracking error by the actor's overall positive progress to the stationary policy, which is different from the existing analysis of (stochastic) PG-type algorithms that mainly decouples or asymptotically decouples the critic's error from actor's error. Further, we develop a new analysis to bound the estimation error of the Fisher information of deterministic policy arising via the compatibility theorem, and then further capture how such a metric affects the convergence via its minimum eigenvalue.

The main proof consists of three steps. First, we characterize the error propagation of tracking a dynamic critic target (i.e., dynamic tracking error) based on its coupling with actor's update progress. Second, we bound the critic's cumulative tracking error in terms of actor's update progress via the compatibility properties of DPG. Last, we establish the overall convergence by canceling out the cumulative tracking error via the actor's overall positive progress towards the stationary policy.

**Step I: Characterizing dynamics of critic's error via coupling with actor.**

In the first step, we characterize the propagation of the dynamics of critic's dynamic tracking error based on its coupling with actor's updates. That is, we develop the relationship between $\left\| w_{t+1} - w^*_{\theta_{t+1}} \right\|^2$ and $\left\| w_t - w^*_{\theta_t} \right\|^2$ by their coupling with actor's updates.

Recall that $w^*_{\theta_t}$ is the global optimum of TD given a fixed policy $\mu_{\theta_t}$, or is equivalently the unique root of $\bar{g}_{\theta_t}(w_t) := \mathbb{E}_{d_{\theta_t}} \left[ \frac{1}{M} \sum_{j=0}^{M-1} \delta_{t,j} \phi(x_{t,j}) \right] = 0$. We first give the following bound on the update rule of $w_t$ in Algorithm 1 given by the TD learning property [Tsitsiklis and Van Roy, 1997,

Bhandari et al., 2018, Xiong et al., 2020],

$$\mathbb{E} \left\| w_{t+1} - w^*_{\theta_t} \right\|^2$$
$$\leq \left( 1 - \frac{\alpha_w \lambda}{2} \right) \mathbb{E} \left\| w_t - w^*_{\theta_t} \right\|^2 + \frac{24\alpha_w^2 (C_A^2 C_w^2 + C_b^2)}{M},$$

where $\alpha_w \leq \frac{\lambda}{2C_A^2}, M \geq \frac{48\alpha_w C_A^2}{\lambda}$.

In the previous analysis of (stochastic) AC algorithms, sufficient TD updates of critic result in a controlled small tracking error before updating the actor, which is hence decoupled from the actor's progress. In contrast, DPG-TD takes alternative updates between critic and actor, so that the critic's tracking error is inherent and non-vanishing. Thus, we take a new approach to characterize the moving dynamics of the tracking error and directly couple it with the actor's update as follows,

$$\mathbb{E} \left\| w_{t+1} - w^*_{\theta_{t+1}} \right\|^2 \leq \frac{4L_w^2}{\lambda \alpha_w} \mathbb{E} \left\| \theta_{t+1} - \theta_t \right\|^2$$
$$+ \left( 1 - \frac{\lambda \alpha_w}{4} \right) \mathbb{E} \left\| w_t - w^*_{\theta_t} \right\|^2 + \frac{48\alpha_w^2 (C_A^2 C_w^2 + C_b^2)}{M}.$$

Clearly, in the above bound, the two tracking errors at times $t+1$ and $t$ have different targets $w^*_{\theta_{t+1}}$ and $w^*_{\theta_t}$ due to actor's one update between critic's two consecutive updates. Hence, actor's update is necessarily coupled into the dynamics of the critic's tracking error.

**Step II: Bounding cumulative tracking error via compatibility theorem for DPG.**

In this step, we bound the cumulative tracking error based on the dynamics of the tracking error from the last step.

To this end, we first bound the difference between two consecutive actor parameters via DPG's properties. By the update rule of $\theta_t$ in Algorithm 1, we have $\theta_{t+1} - \theta_t = \frac{\alpha_\theta}{M} \sum_{j=0}^{M-1} \nabla_\theta \mu_{\theta_t}(s'_{t,j}) \nabla_\theta \mu_{\theta_t}(s'_{t,j})^T w_t := \alpha_\theta h_{\theta_t}(w_t, \mathcal{B}_t)$. Since $h_{\theta_t}(w_t, \mathcal{B}_t)$ is not an unbiased estimator of the deterministic policy gradient $\nabla J(\theta_t)$, we characterize such a bias by exploiting the compatibility theorem as well as the property of Fisher information of deterministic policy defined in Assumption 1 and obtain the following bound (see Lemma 6 for the proof)

$$\mathbb{E} \left\| h_{\theta_t}(w_t, \mathcal{B}_t) - \nabla J(\theta_t) \right\|^2$$
$$\leq 3L_h^2 \mathbb{E} \left\| w_t - w^*_{\theta_t} \right\|^2 + 3L_h^2 \kappa^2 + \frac{6L_\mu^4 C_{w_\xi}^2}{M}.$$

The above bound then connects the critic's error dynamics from Step I to the policy gradient and yields the following result:

$$\mathbb{E} \left\| w_{t+1} - w^*_{\theta_{t+1}} \right\|^2 \leq \left( 1 - \frac{\lambda \alpha_w}{8} \right) \mathbb{E} \left\| w_t - w^*_{\theta_t} \right\|^2$$
$$+ \frac{48\alpha_w^2 (C_A^2 C_w^2 + C_b^2)}{M} + \frac{8L_w^2 \alpha_\theta^2}{\lambda \alpha_w} \mathbb{E} \left\| \nabla J(\theta_t) \right\|^2$$

$$+ \frac{8L_w^2\alpha_\theta^2}{\lambda\alpha_w}\left(3L_h^2\kappa^2 + \frac{6L_\mu^4 C_{w_\xi}^2}{M}\right),$$

where it requires $\alpha_\theta \leq \frac{\lambda\alpha_w}{\sqrt{96}L_h L_w}$.

Thus, we obtain the cumulative dynamic tracking error as

$$\sum_{t=0}^{T-1}\mathbb{E}\left\|w_t - w_{\theta_t}^*\right\|^2 \leq \frac{8\left\|w_0 - w_{\theta_0}^*\right\|^2}{\lambda\alpha_w}$$

$$+\left[\frac{48\alpha_w^2(C_A^2 C_w^2 + C_b^2)}{M} + \frac{8L_w^2\alpha_\theta^2}{\lambda\alpha_w}\left(3L_h^2\kappa^2 + \frac{6L_\mu^4 C_{w_\xi}^2}{M}\right)\right]$$

$$\cdot\frac{8T}{\lambda\alpha_w} + \frac{64L_w^2\alpha_\theta^2}{\lambda^2\alpha_w^2}\sum_{t=0}^{T-1}\mathbb{E}\left\|\nabla J(\theta_t)\right\|^2.$$

The above bound connects the cumulative dynamic tracking error to the convergence rate of actor's update via policy gradient, i.e., such an error depends on how fast actor's update approaches to the stationary point.

**Step III: Overall convergence by canceling tracking error via actor's positive progress.**

In this step, we establish the overall convergence to a stationary policy by novel cancellation of the above cumulative tracking error via actor's update progress.

We first bound the cumulative policy gradient by the cumulative tracking error via the relationship between the progress of loss function and the tracking error as follows:

$$\frac{\alpha_\theta}{4}\sum_{t=0}^{T-1}\mathbb{E}\left\|\nabla J(\theta_t)\right\|^2 \leq \frac{9\alpha_\theta L_h^2}{4}\sum_{t=0}^{T-1}\mathbb{E}\left\|w_t - w_{\theta_t}^*\right\|^2$$

$$+ \frac{R_{\max}}{1-\gamma} + \frac{3\alpha_\theta}{4}\left(3L_h^2\kappa^2 + \frac{6L_\mu^4 C_{w_\xi}^2}{M}\right)\cdot T.$$

The previous analysis of (stochastic) AC typically exploits the fact that the above critic's tracking error can decay sufficiently fast by decoupling it from actor's update, which does not hold here. In contrast, we exploit the connection of the cumulative tracking errors and the cumulative policy gradient that we establish in Step II, and show that such a tracking error can ultimately be canceled by the actor's positive progress towards a stationary point. This also explains why the critic's inaccurate estimation does not affect the overall convergence guarantee. Such an idea is captured as follows:

$$\left(\frac{\alpha_\theta}{4} - \frac{144L_h^2 L_w^2\alpha_\theta^3}{\lambda^2\alpha_w^2}\right)\sum_{t=0}^{T-1}\mathbb{E}\left\|\nabla J(\theta_t)\right\|^2$$

$$\leq \frac{R_{\max}}{1-\gamma} + \frac{18\alpha_\theta L_h^2}{\lambda\alpha_w}\left\|w_0 - w_{\theta_0}^*\right\|^2$$

$$+\left[\frac{48\alpha_w^2(C_A^2 C_w^2 + C_b^2)}{M} + \frac{8L_w^2\alpha_\theta^2}{\lambda\alpha_w}\left(3L_h^2\kappa^2 + \frac{6L_\mu^4 C_{w_\xi}^2}{M}\right)\right]$$

$$\cdot\frac{18\alpha_\theta L_h^2 T}{\lambda\alpha_w} + \frac{3\alpha_\theta}{4}\left(3L_h^2\kappa^2 + \frac{6L_\mu^4 C_{w_\xi}^2}{M}\right)\cdot T.$$

Finally, by letting $\alpha_\theta \leq \frac{\lambda\alpha_w}{24L_h L_w}$ and rearranging the above terms, we complete the proof.

# 4 OFF-POLICY DPG ALGORITHM

In this section, we consider the off-policy setting, where the behavior policy is different from the target policy, and provide the convergence guarantee of such an off-policy DPG algorithm.

## 4.1 ALGORITHM

The design of the algorithm is based on the compatibility property in (5). However, off-policy TD with linear function approximation is known to not necessarily converge [Baird, 1995]. To overcome such a divergence issue, Silver et al. [2014] adopted TD with gradient correction (TDC) to update the critic parameter. Note that since critic in DPG estimates the Q-function rather than the value function, there is no need to use the importance sampling to adjust the sampling distribution [Silver et al., 2014]. We call the compatible DPG using TDC updates as DPG-TDC, with its details given in Algorithm 2.

---
**Algorithm 2** DPG-TDC
---
1: **Input:** $\alpha_w, \eta, \alpha_\theta, w_0, u_0, \theta_0$, batch size $M$, behavior policy $\beta$.
2: **for** $t = 0, 1, \ldots, T$ **do**
3:     **for** $j = 0, 1, \ldots, M - 1$ **do**
4:         Sample $x_{t,j} := (s_{t,j}, a_{t,j}) \sim d_\beta$.
5:         Sample $s_{t+1,j} \sim P(\cdot|s_{t,j}, a_{t,j})$ and $r_{t,j}$. Generate $a_{t+1,j} = \mu_{\theta_t}(s_{t+1,j})$.
6:         Denote $\phi_{t,j} = \phi(s_{t,j}, a_{t,j})$.
7:         $\delta_{t,j} = r_{t,j} + \gamma\phi_{t+1,j}^T w_t - \phi_{t,j}^T w_t$.
8:     **end for**
9:     $w_{t+1} = w_t + \frac{\alpha_w}{M}\sum_{j=0}^{M-1}\left[\delta_{t,j}\phi_{t,j} - \gamma\phi_{t+1,j}\phi_{t,j}^T u_t\right]$.
10:    $u_{t+1} = u_t + \frac{\eta\alpha_w}{M}\sum_{j=0}^{M-1}\left[\delta_{t,j}\phi_{t,j} - \phi_{t,j}\phi_{t,j}^T u_t\right]$.
11:    **for** $j = 0, 1, \ldots, M - 1$ **do**
12:       Sample $s'_{t,j} \sim \nu_{\theta_t}$.
13:    **end for**
14:    $\theta_{t+1} = \theta_t + \frac{\alpha_\theta}{M}\sum_{j=0}^{M-1}\nabla_\theta\mu_{\theta_t}(s'_{t,j})\nabla_\theta\mu_{\theta_t}(s'_{t,j})^T w_t$.
15: **end for**
---

As shown in Algorithm 2, while the state-action pair at time $t$ is sampled by the stationary distribution with the behavior policy $\beta$, the action corresponding to state $s_{t+1}$ is still generated by the target policy. In addition to the updates of $w_t$ as critic and $\theta_t$ as actor, a gradient correction parameter $u_t$ is also updated in line 10 of Algorithm 2.

## 4.2 CONVERGENCE RESULT

In this subsection, we characterize the convergence rate and sample complexity for DPG-TDC in Algorithm 2. Similarly to the analysis of DPG-TD, we first show the Lipschitz continuity property for off-policy DPG-TDC.

**Lemma 2.** *Suppose Assumptions 1-3 hold. Then the deterministic policy gradient* $\nabla J_\beta(\theta)$ *defined in (4) is Lipschitz continuous with the parameter* $L_{J_\beta}$, *i.e.,* $\forall \theta_1, \theta_2 \in \mathbb{R}^d$,

$$\|\nabla J_\beta(\theta_1) - \nabla J_\beta(\theta_2)\| \le L_{J_\beta} \|\theta_1 - \theta_2\|, \quad (8)$$

*where* $L_{J_\beta} = \frac{L_\mu}{1-\gamma}\left(L_Q L_\mu + \frac{1}{2}\gamma L_P^2 R_{\max} L_\mu C_\nu + \frac{\gamma L_P L_r L_\mu}{1-\gamma}\right) + \frac{L_\psi}{1-\gamma}\left(L_r + \frac{\gamma R_{\max} L_P}{1-\gamma}\right).$

Compared with Lemma 1 for the on-policy case, the Lipschitz parameter $L_{J_\beta}$ in Lemma 2 has the same dependence on $1-\gamma$ as $L_J$, but does not have the state visitation error term because the behavior policy does not change as actor updates the policy.

To analyze the off-policy algorithm DPG-TDC, one of the main challenges lies in the complication arising due to the extra correction parameter $u_t$. To overcome this, we treat the update of critic as a lifted linear system with respect to a grouped state $z_t = [w_t^T u_t^T]^T \in \mathbb{R}^{2d}$ and then analyze the key properties of such a system matrix.

We next provide the convergence guarantee for DPG-TDC in the following theorem. The full proof of Theorem 2 can be found in Section 4 of the appendix.

**Theorem 2.** *Suppose that Assumptions 1-4 hold. Let* $\alpha_w \le \frac{\lambda'}{2C_G^2}; M \ge \frac{48\alpha_w C_G^2}{\lambda'}; \alpha_\theta \le \min\left\{\frac{1}{4L_{J_\beta}}, \frac{\lambda'\alpha_w}{24L_h L_{w'}}\right\}; \eta > \max\left\{0, \sigma_{\min}\left(D^{-1} \cdot \frac{A+A^T}{2}\right)\right\}$ *where* $A, D$ *are defined in Assumption 4. Then the output of DPG-TDC in Algorithm 2 satisfies*

$$\min_{t\in[T]} \mathbb{E}\|\nabla J_\beta(\theta_t)\|^2 \le \frac{c_4}{T} + \frac{c_5}{M} + c_6\kappa^2,$$

*where* $c_4 = \frac{8R_{\max}}{\alpha_\theta(1-\gamma)} + \frac{144L_h^2}{\lambda'\alpha_w}\|z_0 - z_{\theta_0}^*\|^2, c_5 = \left[48\alpha_w^2(C_G^2 C_w^2 + C_\ell^2) + \frac{48L_{w'}^2 L_\mu^4 C_{w_\xi}^2 \alpha_\theta^2}{\lambda'\alpha_w}\right] \cdot \frac{144L_h^2}{\lambda'\alpha_w} + 36L_\mu^4 C_{w_\xi}^2, c_6 = 18L_h^2 + \frac{24L_{w'}^2 L_h^2 \alpha_\theta^2}{\lambda'\alpha_w}$ *with* $C_G^2 = 5(1+\eta^2)C_\phi^4, C_\ell^2 = (1+\eta^2)R_{\max}^2 C_\phi^2, L_{w'} = \frac{L_{J_\beta}}{\lambda_\Psi} + \frac{2L_\mu^2 L_\psi C_Q}{\lambda_\Psi^2(1-\gamma)^2}, \kappa$ *given by (7),* $L_{J_\beta}$ *defined in Lemma 2, and other constants remain the same as those in Theorem 1.*

Theorem 2 can readily imply the sample complexity for the convergence of DPG-TDC as given below.

**Corollary 3.** *Suppose the conditions in Theorem 2 still hold. Then the output of DPG-TDC in Algorithm 2 satisfies*

$$\min_{t\in[T]} \mathbb{E}\|\nabla J_\beta(\theta_t)\|^2 \le \epsilon + c_6\kappa^2,$$

*by using the total number of samples* $2MT = \mathcal{O}\left(1/\epsilon^2\right).$

In Corollary 3, the system error $\kappa$ is determined by the off-policy distributions, and thus differs from that of the on-policy DPG-TD algorithm. Overall, Corollary 3 shows that off-policy DPG-TDC achieves the same sample complexity as on-policy DPG-TD in Corollary 1 (up to a different system error). To our best knowledge, there has been no existing study on off-policy *stochastic* AC, where critic uses TDC with general nonconvex policy function approximation. Our techniques can be extended to fill such a gap.

## 5 CONCLUSION

This paper provides the first finite-sample analysis for DPG algorithms in both on-policy (DPG-TD) and off-policy (DPG-TDC) settings. Up to the system error that necessarily exists for actor-critic algorithms, we show that both DPG-TD and DPG-TDC can find an $\epsilon$-accurate stationary policy with a sample complexity of $\mathcal{O}(\epsilon^{-2})$. Our results and the analysis techniques can lead to several promising extension directions. For example, it would be important to explore whether DPG converges to a globally optimal policy as stochastic PG/NPG. Convergence of more popular algorithms such as DDPG is also interesting to study.

### Acknowledgements

The work of H. Xiong, T. Xu and Y. Liang was supported in part by U.S. National Science Foundation under the grants CCF-1761506 and CCF-1900145. The work of L. Zhao was supported in part by the Singapore Ministry of Education Academic Research Fund Tier 1 under the grant R-263-000-E60-133. The work of W. Zhang is supported in part by the National Natural Science Foundation of China under Grants 62073159 and 62003155, the Shenzhen Science and Technology Program under Grant JCYJ20200109141601708, the Science, Technology and Innovation Commission of Shenzhen Municipality under grant ZDSYS20200811143601004.

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
