# OpenReview forum: "Deterministic Policy Gradient: Convergence Analysis"
_auai.org/UAI/2022/Conference — UAI 2022 Poster_

### Official Review · Reviewer_DZXr · 2022-04-09

**Q2(1) Originality/Novelty:** 3
**Q2(2) Significance/Impact:** 3
**Q2(3) Correctness/Technical Quality:** 3
**Q2(6) Clarity Of Writing:** 3
**Q6 Overall Score:** 8
**Q8 Confidence In Your Score:** 3

**Q1 Summary And Contributions:**

This paper provides the first finite-sample analysis for DPG algorithms in both on-policy (DPG-TD) and off-policy (DPG-TDC) settings. Both algorithms are able to find an $\epsilon$-accuracy stationary policy with a sample complexity of O($\epsilon$^(-2)). Moreover, the authors establish the convergence rate for DPG under Gaussian noise exploration. This paper further develops new analysis tools to address the unique challenges arising due to deterministic policies.

**Q2 Assessment Of The Paper:**

More detailed information regarding each of these aspects is given below:

**Q2(5) Reproducibility:**

3: Good: Key resources (e.g., proofs, code, data) are available and key details (e.g., proofs, experimental setup) are sufficiently well-described for competent researchers to confidently reproduce the main results.

**Q3 Main Strengths:**

1. First finite-sample convergence analysis for on-policy and off-policy DPG algorithms.
2. Develop new analysis tools to address the unique challenges for deterministic policies.
3. This paper finds in theory that the convergence rate for DPG under Gaussian noise exploration is similar to DPG.

**Q4 Main Weakness:**

This is a very strong theory paper.  I think it is totally fine without an experiment section. It might strengthen the paper if the author could provide some experiments illustrating the sample complexity O($\epsilon$^(-2)) w.r.t. the $\epsilon$-accuracy.

**Q5 Detailed Comments To The Authors:**

This is a very strong theory paper. It is good to discuss the proof sketch in Sec 3.4. However, it might be better to take less space in the main paper or move them into Appendix?

**Q7 Justification For Your Score:**

This is the first paper with finite-sample convergence analysis for on-policy and off-policy DPG algorithms, achieving the convergence rate of O($\epsilon$^(-2)). These results imply that DPG is as efficient as Policy Gradient with the same complexity. Moreover, the paper establishes the same convergence rate for DPG under Gaussian noise exploration. The authors further develop analysis tools to address the challenges from deterministic policies.

**Q9 Complying With Reviewing Instructions:**

1: Yes.

---

### Official Review · Reviewer_uTth · 2022-04-13

**Q2(1) Originality/Novelty:** 2
**Q2(2) Significance/Impact:** 2
**Q2(3) Correctness/Technical Quality:** 3
**Q2(6) Clarity Of Writing:** 3
**Q6 Overall Score:** 6
**Q8 Confidence In Your Score:** 4

**Q1 Summary And Contributions:**

This paper studies the convergence rate of deterministic policy gradient algorithms. The authors follow similar analysis techniques in previous papers for stochastic policy gradient algorithms and showed that even though the gradient estimator is different, DPG still converges to the stationary point with the same sample complexity as PG with stochastic policies. They further extended their analysis to DPG in the off-policy setting and yielded similar results.




**Q2 Assessment Of The Paper:**

More detailed information regarding each of these aspects is given below:

**Q2(5) Reproducibility:**

3: Good: Key resources (e.g., proofs, code, data) are available and key details (e.g., proofs, experimental setup) are sufficiently well-described for competent researchers to confidently reproduce the main results.

**Q3 Main Strengths:**

The writing is clear. The theoretical results are new in the related literature.

**Q4 Main Weakness:**

The proof techniques are similar to that used in stochastic policy gradients. There are not many new insights into the proofs.

**Q5 Detailed Comments To The Authors:**

The paper is well written. The theoretical results are solid and it is interesting to see such results for DPG which is less studied in the literature on theoretical RL. One potential weakness is its technical novelty is not strong compared with existing work on stochastic policy gradients.

The results presented in Theorem 1 and Theorem 2 are too complicated. There are a lot of problem-dependent parameters that are hidden in the constant c. For example, 1/(1-\gamma) could be large for small \gamma and thus should be shown explicitly in the convergence and the complexity results.

For the off-policy results, it is important to make it clear in Section 4.1 that the convergence is with respect to $\nabla J_{\beta}$ instead of the true performance function. It should be discussed how large can one expect the distance between $\nabla J_{\beta}$ and $\nabla J$ to be.

**Q7 Justification For Your Score:**

Overall, I think the results for deterministic policy gradient are new and outweigh its weakness.

**Q9 Complying With Reviewing Instructions:**

1: Yes.

---

### Official Review · Reviewer_ad3j · 2022-04-13

**Q2(1) Originality/Novelty:** 3
**Q2(2) Significance/Impact:** 3
**Q2(3) Correctness/Technical Quality:** 3
**Q2(6) Clarity Of Writing:** 4
**Q6 Overall Score:** 7
**Q8 Confidence In Your Score:** 3

**Q1 Summary And Contributions:**

The paper provides the first finite-time bounds on the convergence of DPG proposed in Silver et al. (2014).

**Q2 Assessment Of The Paper:**

More detailed information regarding each of these aspects is given below:

**Q2(5) Reproducibility:**

3: Good: Key resources (e.g., proofs, code, data) are available and key details (e.g., proofs, experimental setup) are sufficiently well-described for competent researchers to confidently reproduce the main results.

**Q3 Main Strengths:**

The paper provides new interesting theoretic results on the DGP algorithm proposed by Silver 2014

**Q4 Main Weakness:**

I do not find huge weekness.

**Q5 Detailed Comments To The Authors:**

The paper presents the new interesting results. The paper is well written.

**Q7 Justification For Your Score:**

The paper provides nice new results

**Q9 Complying With Reviewing Instructions:**

1: Yes.

---

### Official Review · Reviewer_DT7R · 2022-04-13

**Q2(1) Originality/Novelty:** 3
**Q2(2) Significance/Impact:** 3
**Q2(3) Correctness/Technical Quality:** 3
**Q2(6) Clarity Of Writing:** 3
**Q6 Overall Score:** 6
**Q8 Confidence In Your Score:** 3

**Q1 Summary And Contributions:**

This is a theoretical work. The author(s) analyzed the non-asymptotic convergence rate of DPG, which is claimed to be the first non-asymptotic convergence of DPG methods.

**Q2 Assessment Of The Paper:**

More detailed information regarding each of these aspects is given below:

**Q2(4) Quality Of Experiments (Optional):**

3: Good: The experimental evaluation is adequate, and the results convincingly support the main claims.

**Q2(5) Reproducibility:**

3: Good: Key resources (e.g., proofs, code, data) are available and key details (e.g., proofs, experimental setup) are sufficiently well-described for competent researchers to confidently reproduce the main results.

**Q3 Main Strengths:**

- The author(s) studied the convergence rate of DPG, which is a descent contribution.
- The author(s) gives the first convergence characterization of DPG method.

**Q4 Main Weakness:**

- Some statement are not accurate enough.


**Q5 Detailed Comments To The Authors:**

- In the abstract and contribution section. The author(s) stated that they derive a sample complexity of $O(\epsilon^{-2})$ to stationary point. But this statement is not accurate. According to Theorem 1 and Theorem 2, the error does not goes to 0 and there is a system error $\kappa$. A more accurate statement should be: derive a sample complexity of $O(\epsilon^{-2})$ to stationary point up to a system error.
- There should be more (and careful) discussion about the system error. It is unavoidable?
- From the optimization theory, usually the variance term is controlled by diminishing learning rate but not a large batch size, is it possible to use fixed batch size but diminishing learning rate to obtain similar result? More precisely, $M$ independent from $\epsilon$ but learning rate depend on $\epsilon$.

Minor:
- First sentence in Introduction, use \citep.
- Proposition, A function Q is said to be -> A function Q is …, “is said to be” sounds like a definition.

**Q7 Justification For Your Score:**

I am not very familiar with the literature of DPG. I vote for acceptence if this work is indeed the first non-asymptotic rate for DPG.

**Q9 Complying With Reviewing Instructions:**

1: Yes.

---

### Decision · Program_Chairs · 2022-05-15

**Decision:**

Accept (Poster)

**Comment:**

Meta Review: This paper provides a convergence Analysis for deterministic policy gradient. The reviewers are generally very positive about this paper, and there is unanimous support to accept this paper. Thus, I recommend acceptance.